# Social Relationships of Captive Bachelor Przewalski’s Horses and Their Effect on Daily Activity and Space Use

**DOI:** 10.3390/ani15010053

**Published:** 2024-12-28

**Authors:** Anastasiia Nykonenko, Yevhen Moturnak, Philip Dunstan McLoughlin

**Affiliations:** 1Department of Biology, University of Saskatchewan, 112 Science Place, Saskatoon, SK S7N 5E2, Canada; anastasiia.nykonenko@usask.ca; 2Faculty of Economics, Business and International Relations, University of Customs and Finance, 2/4 Volodymyra Vernadskoho St., 49000 Dnipro, Dnipropetrovsk Oblast, Ukraine; eugene.mev@gmail.com

**Keywords:** animal welfare, social behaviour, Przewalski’s horse, takhi, threatened species, captivity, sociality, time budgets, behavioural synchrony, enclosure utilization, social network analysis

## Abstract

Studying social relationships and their effects on access to resources of endangered species maintained in captivity is important for ensuring well-being and success of potential releases into the wild. Here, we detail the relationships and daily activities, including space use, of endangered Przewalski’s horses kept in a zoo setting to better understand how to meet the needs of older male (bachelor) horses, an understudied stratum of sociobiology. In our study, we observed nine males (aged 6–21 years) housed together at the Askania-Nova Biosphere Reserve, Ukraine, collecting 65 h of behavioural data. We found that these males naturally split into three distinct subgroups based on their closeness in space and social interactions. Importantly, these subgroups behaved differently when it came to feeding, socializing, and moving but similarly when it came to devoting time to resting and being alert. Horses from different subgroups synchronized their behaviour and used key locations and, hence, resources of hay and water unequally. Our approach may help identify socially important traits, e.g., regarding the most suitable individuals for future releases into the wild, while also pointing out socially driven differences in access to resources that may be acted upon by managers to ensure equity in access to resources.

## 1. Introduction

Developing a thorough understanding of the social behaviour of group-living species in captivity is critical for facilitating conservation efforts [1]. The ability to exhibit natural social behaviours like the expression of social preferences is a key aspect of animal welfare and important for reproduction and productivity [1,2,3]. For example, the formation of enduring social bonds is known to enhance fitness in female feral horses (*Equus ferus caballus*) [4], free-ranging chacma baboons (*Papio ursinus*) [5,6,7], and wild savannah baboons (*Papio cynocephalus*) [8]. Inadequate social conditions and/or suppressed social status may impair fitness in captivity, which can be detrimental to reintroduction programs and conservation of genetic diversity of any species [1,9].

The Przewalski’s horse or takhi (*E. ferus przewalskii* Poljakov, 1881) is well known for its remarkable history of survival in captivity, having descended from < 20 founder individuals after complete extinction of the wild population by the 1960s [10,11]. Moreover, with reintroduction projects in Mongolia and China Przewalski’s horses returned to their natural habitat [12,13,14,15,16,17,18,19,20,21,22]. Some horses were released into conservation reserves across Eurasia (Holland, England, France, Hungary, Ukraine, Uzbekistan, and the Russian Federation) [10,21,23,24,25,26]. These conservation efforts led to a change in the conservation status of the species from Extinct in the Wild (EW) to Endangered (EN) by the International Union for the Conservation of Nature (IUCN) in 2011 [10,21]. By 2020, the global population increased to almost 2500 individuals [21,27]. However, many Przewalski’s horses continue to be bred in captivity to maintain genetic diversity and to provide a source population for augmentation and reintroduction programs [10,11].

Studying the sociality of species like the horse is critical for providing appropriate husbandry conditions and ensuring animal welfare but also for identifying the most suitable individuals for potential release into the wild, given that post-release they must communicate effectively with conspecifics [24]. In the context of reintroduction, social skills acquired in captive settings have proven critical to fitness in the wild including for primates, rodents, and carnivores [28,29,30]. For example, black-tailed prairie dogs (*Cynomys ludovicianus*) reintroduced in family groups had higher survival and reproductive success than individuals reintroduced without regard to family group [28].

Social structure in equids varies from solitary territorial species that can also form fusion-fission societies like Grevy’s zebra (*E. grevyi*) and African wild ass (*E. africanus*) to coherent family groups in feral horses, plains zebra (*E. quagga*), and mountain zebras (*E. zebra*) [10,31,32,33,34]. The three latter species exhibit female-defense polygyny, forming breeding groups called bands [31,35,36,37]. A stallion that acquires a harem of females defends them (and their offspring) from other males throughout the year (with competition somewhat relaxed in winter) [37]. Juveniles remain with the natal band until reaching sexual maturity, typically at 2–3 years of age for females (fillies) and 3–4 years for males (colts) [36,38,39]. Dispersing immature males and stallions that fail in obtaining and retaining females rarely exist in isolation; rather, it is typical for them to aggregate into bachelor groups [31,38,39,40,41]. Even though the social structure of Przewalski’s horses was not well known before the species went extinct in the wild [42], it is assumed that they also had a harem structure similar to the above [43,44,45,46]. Reintroduced Przewalski’s horses form harems and bachelor groups, and sometimes multiple breeding groups aggregate into a larger herd [14,15,17,19,20,27].

Social integration is therefore particularly important for horses, where stallions compete for access to females. Stallions with poor competitive skills may be excluded from breeding, thus failing to contribute their genetic material to the population, which could jeopardize species or group persistence probability [13]. In the context of male equids, time spent in non-natal but non-breeding groups of other males (bachelor groups) is essential for the development of social skills to be used later in life [38]. Young males gain important social skills for securing and defending a future harem as they mature [25,44,46]. Moreover, the social position in these bachelor groups affects reproductive success, as has been shown for stallion Przewalski’s horses with the positive correlation between the dominance rank in the bachelor group with the number of foals sired later in life [47].

The correct development of social behaviour is a crucial aspect for the released groups success in the wild, as stated by IUCN/SSC Equid Specialist Group in the Action Plan for the Przewalski’s horse [48]. While much is known of the social dynamics of horses (including Przewalski’s horses) in the context of band relationships, less is known of the behaviours exhibited among bachelor males [31,36,38,40,49,50]. In part, this is because non-breeding horses in semi-wild and captive conditions are often kept in artificial social constructs [9,42,43,44,45,46]. Even less is known of how older or surplus males (a focus of this study), which are no longer candidates for reintroduction, fare in relation to forced social constructs in captivity.

Maintaining Przewalski’s horses in captivity requires taking many factors into account, including genetic compatibility (if breeding), health conditions, and more [26,42]. It has been shown that >100 years of captive breeding has resulted in reduced heterozygosity, increased inbreeding, and variable introgression of domestic alleles [10]. Maintaining welfare of captive animals also entails considerations of social confinement, including sex and age structure as well as availability and size of enclosures and housing conditions, etc. [42,51]. For equids, a common result is the formation of artificial bands constructed from genetic considerations over behaviour, while surplus males not intended for breeding are left either to live alone or are forced together in bachelor groups. Both housing arrangements come with risks: the former may lead to stereotypic behaviours (e.g., pacing) due to solitary living [42], while the latter can result in increased aggression between stallions forced to tolerate one another in confined spaces [46], a problem also observed in domestic horses [2]. Keepers must carefully examine how social relationships within these artificial groups impact daily life, welfare, fitness, and potential reproductive success [42].

Analysis of behaviour in comparison to an expected benchmark is a simple and effective tool for assessing animal welfare in captivity [3,46,52]. For example, data on time budgets (qualitative and quantitative) and evenness of enclosure use can be collected to detect abnormalities in behaviour that could have negative fitness consequences [53,54,55]. Time budgets are widely used to evaluate the well-being of domestic horses [52] and various zoo-housed species [56]. Social covariates considered in relation to time budgets include sex, reproductive status, and dominance rank [57], while factors like gregariousness, individual centrality, and subgroup formation are often overlooked [58].

Social structure of a group consists of social relationships built on interactions and associations [59]. Studying associations and interactions in animal groups can therefore provide insights into their social structure and its influence on daily life [28,51,55,60]. The aim of this study was to describe the social structure for a sample (captive) bachelor group of older Przewalski’s horses (ages 6–21 years), assess their time budgets and space use with different space availabilities, and investigate how social relationships influenced time allocation and territory utilization. While our focus is on the sociobiology of older males, our results likely apply to the housing of male horses in general. The first hypothesis we aimed to test was that associations and interactions were not random due to individual preferences in social partners. We predicted that each individual would have a preferred social partner in spatial associations and affiliative interactions, and both to be correlated. Additionally, we predicted that the formation of associations and affiliative bonds (from friendly interactions) would emerge from higher genetic relatedness, similar age, dominance index, and gregariousness.

Next, we described typical time budgets throughout the day. We predicted feeding and rest to be the primary activities of all horses, but rates of grazing and foraging to be different among animals due to preferential access to high-quality hay in high-ranking individuals. We also hypothesized that with the availability of a second enclosure, such effects would relax as horses would be freer to redistribute as predicted by models like the ideal free distribution [61], which predicts equal distribution of space access per individual in the absence of social constraints on movement. The alternative, deviation from ideal-free distribution, would suggest continued despotism of access to space related to social hierarchy.

Finally, we hypothesized that social preferences would affect time budgets and space use because social partners would be synchronizing their behaviour and use the territory similarly, particularly in relation to key resources (hay and water) within the enclosure. We predicted that horses forming associations and frequently interacting would have more similar time budgets and demonstrate behavioural synchrony. We also predicted that high-ranking individuals would enjoy more rest time and require less feeding time, as they monopolized high-quality resources and therefore needed less time foraging. In terms of space use, we predicted that high-ranking individuals would monopolize important resources and use areas with hay and water more than lower-ranking individuals.

## 2. Materials and Methods

We studied the social behaviours of adult, bachelor Przewalski’s horses at the Askania-Nova Biosphere Reserve, for 25 days from 21 July to 14 August 2015. Located in the Kherson region of southern Ukraine, the reserve is one of the oldest and most prominent nature reserves in Europe and was designated a UNESCO Biosphere Reserve in 1985 [62]. Spanning approximately 33,000 hectares, Askania-Nova is composed of three primary areas: the pristine steppe, the arboretum, and a zoo. The steppe constitutes over half the reserve area, representing one of the last remaining sections of undisturbed temperate grassland in Europe. Its preservation is crucial for maintaining steppe biodiversity and serves as a refuge for native fauna [63]. The zoo and animal park, where our work was conducted, has played a significant role in the breeding and reintroduction of endangered animals including Przewalski’s horse [44,45,63]. Animals are maintained in semi-wild conditions within expansive enclosures that mimic their adjacent natural habitat, allowing for more natural behaviour and successful reproduction [44,46,64].

For this study, we focused our observations on a group of nine adult bachelor Przewalski’s males aged 6–21 years (Table 1). Breeding experience and health status of the studied individuals were unknown (but none had registered foals in the studbook), and body condition was not scored. At the start of our study, the level of familiarity, e.g., how long individuals were in the same group, was also unknown. Horses shared the enclosure with a herd of 18 Mongolian wild ass or khulans (*E. hemionus hemionus*). No visitors were allowed at the enclosure or near it at any time during the period of observation.

During the 3-week intensive observation period the group lived in Enclosure I, with a total area of 3.78 ha (Figure 1). For the first 2 weeks of observation, from 21 July to 3 August, horses also had access to the Enclosure II, 0.84 ha in size. To study space use, enclosures were roughly divided into 22 equal-sized zones/quadrats (30 × 70 m). In addition to evenly distributed grass cover, available for grazing in both enclosures, animals had *ad libitum* access to a prescribed diet of hay with fresh hay added daily (zones A3–A6, B3–B6). Fresh water was available *ad libitum* at a freshwater container (repurposed metal bath approx. 1.5 × 0.5 m in size) in zone C6. No shelter was provided for the animals.

Median daytime air temperature throughout the study was 33 °C (ranged from min 28 °C, max 37 °C) and 25 °C in the evening (min 20 °C, max 29 °C). On most days the weather was sunny with only a couple of cloudy days.

The nine subjects were observed almost every day once or twice a day for a time sample of 2 to 4.5 h during the hours of 0500 h to 2030 h for a total of 65 contact hours. This number of contact hours is justified with a case study on feral and Przewalski’s horses that found 15 h of observations per group to be sufficient for robust and reliable data for the analysis on social bonds [65]. Care was taken to equally distribute the study hours throughout the day (morning: 0500–1000, afternoon: 1010–1500, evening: 1510–2030) during the field season. Only one observer (AN) collected all behavioural data. Observational methods were validated from the previous studies mentioned below; and a 3-day pilot study and individual-horse recognition practice were initiated prior to any data collection. For data collection, the observer always remained > 50 m away from horses, to minimize effects of observer presence on animal behaviour. All individuals were visible and identified during data collection. Horses were recognized by cold brands, earmarks, and fur colour and features. A pair of 7 × 35 field binoculars were used to aid in individual recognition.

Activity patterns were registered using instantaneous scan sampling with an interval time of 10 min for time budget calculations [66]. Activity patterns used in sampling were grouped into the following behavioural categories: foraging, rest, locomotion, social, vigilance, other (Table 2). At the same time as behavioural sampling, the location of each horse within enclosures was recorded to analyze space use. In horses individuals who stay less than two (2) horse body lengths (HBL) apart from one another are usually considered as connected [67]. One HBL (approx. 1.5 m) corresponds to the horse’s individual distance, which is commonly from 1 to 2 m [68]. Horses within 1 HBL from one another were considered associated at a given scan sample. If more than two horses stayed within 1 HBL from a given individual, all were considered associated to one another. Distance from horses to khulans was not measured at any time. Interspecies social interactions were not recorded, either.

All-occurrence sampling was used to record all social interactions between individuals, noting the actor, recipient, and the content of interactions [66]. The ethogram for social interactions followed that of previous studies [44] (Table 3).

All data analyses were performed with R (R Core Team, 2021), using the RStudio interface (version 2023.12.1; RStudio Team, 2023) [69]. The total number of scan samples for all horses used to construct spatial social networks was 3989. The definitions and procedures of social data analysis are presented in the Table 4.

Pedigree trees for all studied individuals dating back to the founders were available from the Studbook of the Przewalski’s Horse [71]. Kinship coefficients using the pedigree were calculated with a function kinship from the *Kinship2* package (Appendix B). Correlations between associations, affiliative interactions, and kinship matrices were assessed with a Mantel test from package *Vegan*.

Node betweenness, closeness, and eigenvector centrality were calculated for associations and affiliative interaction networks but were not included in the consequent analysis because of high correlation with the weighted degree. To quantify individual gregariousness, we used two node measures from association and interactions networks (node strength, weighted degree, see Table 4).

Features of dominance relationships were calculated with a package *compete*. Linearity test for dominance relationships within a band was completed with a function devries (randomization test, controlling for unknown dyadic relationships). David’s scores for all individuals were computed with a function ds. David’s score or dominance index (DI) assesses the relative dominance of individuals based on pairwise comparisons between dyads and reflects the relative number of times the individual wins an aggressive encounter (or triggers a submissive behaviour), thus dominating over the others. Here, we follow the definition of dominance as an attribute of the agonistic interactions between two individuals, with a consistent outcome in favour of the same dyad member (dominant) and a default yielding response of its opponent (subordinate) rather than conflict escalation [72]. While we expect dominance relationships to be present within dyads, that would not imply the existence of the strict linear dominance hierarchy within a studied group [73,74].

We computed a behavioural synchronization index (BSI) as the proportion of time two individuals in a dyad performed the same behaviour (out of all 10-min scan-samples), ranging from 0 to 1 [75]. To account for the probability of horses engaging in the same behaviour purely by chance, a randomization test was performed [76]. In every permutation, a random horse was chosen and the sequence of randomly chosen 100 scan samples was shuffled. The BSI was then calculated for the permuted data in each randomization. A one-tailed *t*-test was then performed to calculate *p*-values (as a proportion of permuted behavioural synchronization indices greater than the observed index).

To calculate the relative use of zones within an enclosure, an electivity index (EI) was calculated for each zone as a ratio between an observed and expected use of it. The formulas for EI calculations are given in Appendix C (Formulae (A2) and (A3)) [77]. EI were computed per each individual zone and vary from −1 (no use of the zone) to 1 (maximum use of that zone) [53]. To calculate the overall evenness of the enclosure usage for each horse, a spread of participation index (SPI) was calculated with a formula given in the Appendix C (Formula (A4)). The SPI accounts for an observed and expected use of each zone within an enclosure, and varies between 0 (maximum enclosure use, all zones used equally) and 1 (minimum use, only one zone used) [78]. Generalized mixed effects models (GLMM) were used to quantify the effect of social position on daily behaviour and space use. Fixed effects included social factor (cluster), time of day, availability of the second enclosure, while the horse identity was treated as a random effect (package *lme4*). A separate model was fitted for a probability of each behaviour (foraging (in addition, models were fitted separately for feeding on hay and grazing), rest, locomotion, social, vigilance) in relation to fixed and random effects with binomial distribution. The same model with hay availability as a response variable was fitted to analyze differences in space use in relation to social factor. Additionally, a model with electivity index as a response variable, all same predictors (and hay availability) and gaussian family was fitted.

## 3. Results

### 3.1. Social Relationships 

#### 3.1.1. Testing Non-Randomness of Associations and Interactions

To assess the association patterns within a bachelor group of Przewalski’s horses, a matrix of AI was computed and a social network based on the matrix values was constructed (Figure 2).

Horses spent from 0.2% to 66.5% of time associated with different individuals (observed mean ± standard deviation (sd) of AI is 0.081 ± 0.159, coefficient of variation (CV) of AI is 1.954, see Table 5). Associations were significantly different from random, as calculated with the difference between CV of AI in dyads within observed and permuted networks (*n* = 1000, *p* < 0.01)

Allogrooming was the most frequent affiliative social behaviour (335/358 of interactions, 93.6%). Rates of affiliative interactions were calculated as the number of affiliative interactions per hour (observed mean ± sd of interaction rates is 0.112 ± 0.264, min = 0.015, max = 1.077, CV of interaction rates = 2.357; Table 6). Direction of friendly interactions and the number of interactions initiated and received in in studied horses were significantly different from random, as calculated with the difference between CV of interaction rates between stallions within observed and permuted interaction networks (*n* = 1000, *p* < 0.01).

#### 3.1.2. Preferred Social Partners 

Subjects clearly had preferred and avoided associates, as quantified from the difference between each dyad AI within observed and permuted networks (Table 5). Each horse had at least one preferred associate with whom they spent significantly more time than expected by chance. All horses also had avoided associates with whom they spent significantly less time than expected by chance (see Table 5). Subjects also had preferred social partners with whom they interacted affectionately significantly more than is expected by chance and consequently avoided social partners (Table 6). The directionality and rates of affiliative interactions within a study group are reflected on Figure 3.

The Louvain community-detection algorithm identified three social units within a group of horses based on proximity associations (see Figure 2). Modularity score from the association matrix was 0.593 (Louvain method, *p* < 0.01, as derived from 1000 permutations). The Infomap algorithm found the same clusters based on affiliative interaction rates with a modularity score 0.350 (see Figure 3).

#### 3.1.3. Correlation Between Affiliative Interactions and Associations

While the same subunits (clusters) of social partners were identified in both associations and affiliative interactions, no significant correlation was found between the matrices of AI and affiliative interactions rates (Mantel correlation coefficient *r* = 0.035, *p* = 0.44). Node measures from association and interaction networks summarized the gregariousness of individuals (Table 7). Cluster membership was used as a social factor in the consequent analysis of daily activity and space use as some similar social tendencies were identified among clusters. Cluster 1 could be characterized as the most associative both with a higher number of spatial associates for each individual (higher WD_assoc_) and stronger bonds quantified as the total time spent associated (higher S_assoc_). In Cluster 2, Losk and Palats were the most associative (both measures higher), while Zakat was less sociable, and Lepet was the least. Cluster 3 was highly connected within itself (high S_assoc_, but low WD_assoc_) (Figure 4).

Considering affiliative interactions, members from Cluster 2 were gregarious both in the number of horses each individual interacted with (higher WD_affil_) and total interactions rate for each individual (higher S_affil_). Horses from Cluster 1 were less interactive with both measures, and Cluster 3 was isolated within itself (both measures lower).

#### 3.1.4. Agonistic Interactions 

Agonistic interactions consisted of aggressive (offensive) and submissive (defensive) behaviours (see Table 2). Rates of agonistic interactions were calculated as the number of agonistic interaction bouts per hour (observed mean ± sd interactions rate is 0.039 ± 0.124, min = 0.015, max = 0.569, CV of interactions rate = 3.188). Most encounters were submissive (65/107, 60.7%) mainly consisting of avoidance (62/65). Aggressive harassment composed most of the offensive encounters (33/42), and rates of other interactions were negligible.

We combined offensive and defensive social behaviours for the network construction with an initiator (whoever initiate the offensive or defensive interaction) and recipient (whoever the offensive or defensive interaction is directed to) (Figure 5). For the dominance hierarchy computation, we treated initiators of offensive interactions as winners. However, we reversed the direction of defensive behaviours so that winners were those to whom submissive behaviours were directed (recipients of submissive interactions). The resulting dominance hierarchy appeared to be non-linear (the modified Landau’s h′ = 0.346, *p* = 0.334). Therefore, the dominance indices were not used to calculate social ranks. However, several individuals were noted to have the most disproportionate fraction of wins and losses: Lepet initiated and won the most aggressive encounters and provoked the most submission (the most dominant horse with the highest DI, see Table 7). In contrast, Lovelas and Vernij were the most frequent losers and showed submission the most frequently (subordinate individuals with the lowest DI, see Table 7).

#### 3.1.5. Assortment in Associations and Affiliative Interactions

Genetic relatedness did not significantly affect spatial associations in the group of bachelors (Mantel correlation coefficient *r* = −0.230, *p* = 0.864). The propensity to interact friendly with a conspecific did not correlate with genetic relatedness either (Mantel correlation coefficient *r* = 0.230, *p* = 0.136).

Assortment by age was not significant in either associations (assortment coefficient, AC = −0.084, *p* = 0.415) or interactions (AC = 0.025, *p* = 0.249). Horses tended to associate and interact with individuals of similar DI (AC = 0.433, *p* = 0.013 and AC = 0.119, *p* = 0.165 correspondingly).

Horses associated with individuals of similar WD_assoc_ (time spent associated corrected for the number of close associates, AC = 0.856, *p* = 0.002), but not of WD_affil_ (rates of interactions corrected for the number of interactants), but interacted regardless (AC = −0.028, *p* = 0.290 and AC = −0.029, *p* = 0.269 correspondingly). Horses associated and interacted regardless total gregariousness derived from time spent associated S_assoc_ (AC = 0.012, *p* = 0.250 and AC = −0.254, *p* = 0.685 respectively). However, horses associated based on total gregariousness derived from affiliative interaction rates S_affil_ (AC = 0.638, *p* = 0.008) and interacted with those interacting a lot (AC = 0.506, *p* = 0.031).

### 3.2. Time Budgets 

Foraging (including both feeding on hay and grazing on grass) was a primary activity of all the horses in the bachelor group (56.52 ± 4.26% on average). Feeding on hay and grazing differed considerably between horses (Appendix D, Table A2). Lovelas (2.70%) and Vernij (2.45%) fed much less compared to the rest of the group (29.34–40.36%). In contrast, they grazed on the grass more (Lovelas 53.81%, and Vernij 52.01% of time) than others (17.34–28.79%). All horses spent almost a quarter of time resting (19.93 ± 7.43%) and stayed vigilant for 10.53 ± 5.17% of time (Figure 6). Locomotion took 8.98 ± 1.98% of time, 2.45 ± 1.99% was devoted to social behaviour and other activity patterns took only 1.58 ± 0.83% of time.

Subjects followed a distinct daily schedule, engaging in different activities at various times throughout the day. Horses tended to forage (regardless feeding on hay or grazing) in the evening more than compared with the afternoon and even less in the morning (for this and other behaviours GLMM *p* < 0.05 for pairwise comparisons between afternoon and other two periods of the day, morning, and evening). Rest was similarly frequent in the morning and afternoon, with a lower likelihood in the evening (see Figure 6). Locomotion took more time in the morning and evening hours compared to the afternoon. Horses demonstrated significantly more social behaviour in the afternoon and significantly less vigilance in the evening.

The availability of the second enclosure significantly affected the probability of foraging, resting, and vigilance behaviour with no effect on locomotion and social behaviour (Appendix D, Figure A1A,B). After the removal of the second enclosure horses increased foraging time (GLMM, Table 8, *z* = 2.838, *p* = 0.005), and vigilance (*z* = 5.878, *p* < 0.01), but reduced resting (z = −6.577, *p* < 0.01).

### 3.3. Space Use 

Subjects used the enclosures mostly unevenly, as identified with a spread of participation index (SPI, Table 9). The mean SPI was 0.453 ± 0.065 [±1. s.d] before the removal of the second enclosure and 0.457 ± 0.098 after removal. SPI did not change significantly after the removal of the second enclosure (ANOVA, *F* = 0.012, *p* = 0.913). The most uneven use of the enclosures was observed in Lepet (the highest SPI = 0.502), while Lovelas and Vernij tended to use the space more evenly (lowest SPI, 0.339 and 0.342 accordingly).

Electivity indices (EI) varied greatly between zones, and zone preferences were different among horses (Appendix E, Figure A2). EI were significantly lower for zones with no important resources (zones A3-A6, B3-B6 with hay and zone C6 with water, see Figure 1; *t* = −8.117, *p* < 0.01).

Horses showed some spatial activity patterns in engaging in specific behaviours in different zones within an enclosure (Appendix E, Figure A3). Zones A1–A2 in the upper corner of the enclosure were used mostly for rest, while feeding on hay prevailed in zones A3-A6 which had most of the hay (see Figure 1). Even though the grass cover was even throughout the enclosure, most of the grazing was observed in zones B1–B2, C1–C3, and in enclosure II (zones D1–D2, E1–E2). When testing for spatial differences in behaviour in relation to hay presence, significantly more rest, locomotion, and vigilance was observed in zones without hay (Table 8, *p* < 0.01). Social behaviour was observed with no regard to hay presence in the given location (z = −0.372, *p* = 0.710). The differences in zone preferences between the periods with and without the second enclosure were not detected (t = 0.766, *p* = 0.444).

### 3.4. Social Status Effect on Time Budgets and Space Use

#### 3.4.1. The Impact of Sociality on Time Budgets

Differences in behaviour of horses belonging to distinct social subunits (clusters) were unequal among activity patterns (Table 8). No significant differences were detected between clusters throughout foraging and rest. However, the members of Cluster 3 were significantly less prone to locomote (*z* = −4.964, *p* < 0.01), remain vigilant (*z* = −2.140, *p* = 0.032), and participate in social behaviour (*z* = −2.792, *p* < 0.01).

Furthermore, when accounting for feeding on hay and grazing, the differences between all three clusters became more apparent with members of Cluster 1 and 2 significantly more probable to feed on hay, while members of Cluster 3 tended to graze more (Table 10).

#### 3.4.2. Behavioural Synchrony

The mean behavioural synchrony index in the bachelor group was not significantly different from random (mean BSI ± sd =0.369 ± 0.167, *p* = 0.158). However, when testing for significant differences between observed and permuted networks on the dyadic level, for some dyads the BSI was higher than expected by chance (Table 11).

Preferred associates showed higher level of behavioural synchrony computed as the mean BSI of all activities (Mantel test correlation between the matrices of AI and BSI, *r* = 0.703, *p* = 0.009).

We further expanded the analysis of behavioural synchrony over each activity (Table 12). Horses showed the highest synchrony when grazing (mean BSI ± s.d. = 0.322 ± 0.185) and feeding on hay (0.316 ± 0.275). Rest and locomotion were synchronized less, and vigilance and social behaviour the least (see Table 12). Different dyads showed various synchrony among activities (Appendix F, Table A3).

Finally, we tested the differences for within- and between-cluster behavioural synchrony for each activity. For all activities, mean BSI was significantly higher in horses within one social cluster than between clusters (see Table 12). The biggest difference in within- and between-cluster synchrony was in feeding on hay (coherent with the results from GLMM) and the least in vigilance. Horses that formed Cluster 1 were highly synchronized in feeding on hay, grazing, and resting (Figure 7). Members of Cluster 2 fed on hay and grazed mostly at the same time as well, while rested less synchronically compared to Cluster 1. Cluster 3 was the most synchronized over all social units in all activities.

#### 3.4.3. The Impact of Sociality on Space Use

Zones of the enclosure with important resources (hay and water) were used differently by horses from different social clusters (Table 8). Members from Cluster 1 were more prone to use zones with hay, while Cluster 2 used them less and Cluster 3 was almost excluded from those zones. Horses spent more time in zones with hay in the afternoon compared with morning and evening and when the second enclosure was not available.

## 4. Discussion

### 4.1. Social Relationships Within a Bachelor Group of Przewalski’s Horses

**Subgroup formation.** Spatial associations and interactions in the studied group of bachelor males of Przewalski’s horses were found to be non-random, as predicted. All horses showed social preferences in both spatial associations and affiliative interactions, organizing themselves in subunits of two (Lovelas and Vernij), three (Bulat, Vitjaz, Parus), and four (Losk, Palats, Zakat, Lepet) and spending up more than half of time together. Clustering of the entire group, the major finding of this study, was significantly different from random and indicated important divisions of the group into subunits (clusters) [52,60,79]. Daily activity patterns and territory usage varied between clusters more than within, as will be discussed further.

Our work, which was comprehensive in terms of social metrics, adds to the few studies that have examined the formation of associations in Przewalski’s horses with a focus on males [25,44,45,46,80,81]. In a broad study across numerous European zoos and reserves proximity in all groups of three to six mixed-aged Przewalski’s bachelors was found to be non-random with preferred and avoided associations [25]. The latter study found significant differences between associations of different-aged individuals with the mean AI between stallions equal 0.29, compared to 0.56 in colts and 0.27 between stallions and colts; the alpha male was always farther from all individuals, while subordinates maintained close proximity to one another [25]. In a study of a mixed-age bachelor group of eight Przewalski’s males in Minnesota, USA, it was found that stable spatial associations formed only between four immature males (70% of time associated), but not between four adults (from 1 to 11% of time associated) [46]. Likewise, in a study of four juvenile males there were clear spatial associations between individuals with one individual being the most central (being nearest to any of the horse frequently) [24]. Similarly, another study of larger male group (*n* = 13) consisting mostly of juveniles kept in the large enclosure in Askania-Nova Biosphere Reserve found the median frequency of having the nearest neighbour at ≤ 1.5 m being 56% [80]. In contrast, we found stable associations between horses even though all bachelors were mature (≥ 13 years) with only one considerably young individual (Zakat, 6 years), indicating some factors other than age contributing to the formation of social bonds.

Researchers, working with a dense Askanian mixed-age group (*n* = 15) of bachelor males, suggested the formation of subgroups based on age-related behaviours: type 1 subgroup being homogenous in members’ behaviour (consisting most of juveniles) and type 2 subgroup with a dominant individual herding other members [44]. These authors believed that the formation of the latter subgroup type was an artefact of the unrealized sexual behaviour with dominant stallions treating other males as females [44]. That could be the case in our study, too. However, with a formation of subgroups only one individual for the entire group was the most dominant over all others (Lepet). Unrealized sexual behaviour as a driver of social dynamics in artificially kept male horses of all ages, including older animals, is an important area of further research.

In the wild, smaller associations of bachelors, as we found, do form naturally. For instance, the mean size of the bachelor group in the population of 270 feral horses of Pryor Mountain Wild Horse Range in the USA is 1.8 individuals with the largest of eight individuals [82]. The latter authors also noted dominant males leading larger groups and behaving similarly to managing a harem. However, often these groups divided into smaller fractions with different dominant males in each, with smaller bachelor groups being more stable [82]. In New Zealand Kaimanawa feral horses, bachelors do not form stable associations but rather enter transient groups of up to 13 individuals [83]. On Sable Island (Canada), bachelor groups of feral horses are also considered to be unstable with the low probability to associate the following year [84]. However, some associations between dyads could last for up to 8 years with associations lasting even after a male acquires a harem [84]. The mean number of bachelors in the group for all horses followed on Sable Island from 2008 to 2023 was 2.09 with a maximum of 12 male individuals observed together at the same time (S. Medill, Sable Island National Park Reserve, pers. comm.).

Contrary to our predictions, neither genetic relatedness nor similarity in age significantly predicted social preferences (notwithstanding Lovelas and Vernij, who were always together, were full brothers). Rather, stallions tended to socialize more with individuals of similar gregariousness and dominance index. Some different social tendencies were identified within observed subunits: in Cluster 1, stallions formed strong associations with more individuals but rarely interacted; in Cluster 2, Losk and Palats formed strong bonds through both associations and interactions, Zakat was interacting even more but with fewer partners and did not associate much, while Lepet was aloof neither associating nor interacting with others. Lovelas and Vernij from Cluster 3 presented a very tight association bond and were behaviourally separated from the rest of the group. Extensive studies of individual attributes, such as personality, individual history, etc., may shed light on why particular individuals are clustered together. The ability to become a dominant male in the bachelor group is usually connected to the individual’s temperament [46]. Age and former experience as a breeding stallion in a harem group could also contribute to the social position. However, in an above-mentioned study of the large bachelor group in the Askania-Nova Biosphere Reserve researchers claimed that these attributes did not influence a stallion’s ability to form a type 2 subgroup with a dominant individual herding other members [44].

**Affiliative interactions, aggression, and dominance hierarchy.** In our study, rates of agonistic encounters were lower than affiliative, which means that the older males that we studied relied more on friendly interactions rather than aggression to maintain social structure. However, contrary to our predictions, associations and affiliative interactions were not correlated. Other researchers also found young Przewalski’s colts to play more upon entering bachelor groups and show almost no aggressive elements [81], and the rates of affiliative interactions to be 0.53 per h per horse [80] and 081 ± 0.68 per h per horse [25] for captive bachelor groups. More friendly interactions than aggressive were also reported in large bachelor group in Askania-Nova Biosphere Reserve (7.71 ± 1.01 and 4.89 ± 0.53 per hour, correspondingly; friendly interactions taking 63% of all the interactions observed) [44]. In the other study of four juvenile males, the proportion of aggressive to non-aggressive encounters was low: 0.31 [24]. Generally, horses associated and interacted less between clusters than within, although Zakat and Lepet showed less consistency in their social bonds.

Primarily, one stallion in our study (Lepet, the second youngest [aged 13 years]) demonstrated most of the aggression in the entire group, which he directed towards a single dyad consisting of Lovelas and Vernij. However, our (constructed) agonistic-encounters dominance hierarchy was not linear, which is probably due to a very low number of encounters between horses apart from the above. Likewise, no linear hierarchy was found in any of the small (*n* from three to six) bachelor groups maintained in semi-reserves and zoos across Europe for the species, also probably due to the low number of recorded interactions and few animals in the groups [25]. However, unlike our work, the latter study found a positive correlation between age and tenure in the group and a social dominance rank [25]. In an early observational study of a large mixed-age and sex herd (*n* > 40) of Przewalski’s horses kept in a 2660 ha enclosure in Askania-Nova Biosphere Reserve, males driven off by the most dominant stallion guarding the harem of females formed a bachelor group “with an internal hierarchy based on a stable non-competitive relationship [until the appearance of reproductive stimuli], and individual stallions, with the development of their physical and psychological status, tried to seize females” [45]. A linear hierarchy was found among fully adult bachelors and the same low rank in four juveniles in the mixed-age Przewalski’s group of eight with one adult male clearly the most dominant [46]. A linear hierarchy was also among four juvenile Przewalski’s males [24]. A linear hierarchy was also found within subgroups of the above-mentioned large bachelor group. However, there was no clear hierarchy in the whole group [44]. In the present study, apart from the aggressive encounters described agonistic interactions were kept to minimum, which served as evidence of well-established social relationships within a group.

### 4.2. Time Budgets and Space Use of Przewalski’s Horses Male Group

**Time budgets.** As predicted, horses spent the most time foraging and resting. Other researchers agree on foraging taking the most part of the time budgets of Przewalski’s horse males: In adult bachelor males kept in zoos, 45.4 ± 5.1% of time is devoted for foraging, with rest taking almost a quarter of time (standing 22.6 ± 2.0 and stand-resting 9.2 ± 2.9%) and locomotion just 16.9 ± 3.8%, while social behaviour took the lowest percentage of time (mutual grooming 0.8 ± 0.4 and play 4.5 ± 1.3%) [85]. That is comparable to the time budgets of reintroduced Przewalski’s horses (harem of nine observed shortly after release at Hustai National Park, Mongolia) that spent most of their time grazing (54.7 ± 0.8%), followed by resting (19.9 ± 0.6%), moving (16.8 ± 0.8%), and standing (7.8 ± 0.7%) [22]. Data from five captive bachelors of Przewalski’s horses kept in summer on a 40 ha pasture showed similar trends: grazing 58.4%, moving 12.2%, standing 3.7%, resting standing 11.0%, and resting recumbent 5.5% [15]. In our study, horses grazed more with the availability of the second enclosure; however, there was no increase in locomotion. Generally, horses rested less and spent more time vigilant after the removal of the second enclosure. This increased alertness might indicate some negative effect of the removal of an additional space on horses’ daily activity.

**Space use.** Our results suggest that on the whole, horses accessed space in an ideal-free manner, as suggested by no change in SPI from when enclosure II and I were available for access. No studies on captive Przewalski’s bachelors were found to compare their space use; however, in the description of spatio-ethological structure of the big herd of Przewalski’s horses in Askania-Nova Biosphere Reserve distribution of the group over the pasture reported to be not random but within a definite spatial structure and configuration: bachelors stayed all together when driven out by a band stallion and not mixing with the entire herd [45]. Generally, zoo animals are known to use space unevenly, preferring areas with important resources. For example, in captive sitatunga (*Tragelaphus spekii*), SPI was identified at a level of 0.61 on average for a group of eight with a significant preference for long-grassy areas of the different in regards of biological relevance zoning of the enclosure [86].

### 4.3. The Effect of Sociality on Time Budgets and Space Use of a Bachelor Group of Przewalski’s Horses

**Sociality effect on time budgets.** According to our predictions, feeding on hay was not equal among the horses with individuals from Cluster 3 (subordinate) feeding on provisioned hay significantly less compared to other horses, but grazing on grass considerably more. Likewise, a study of a captive American bison (*Bison bison*) herd of 11 suggested differential time use in relation to social status with high-ranking individuals spending significantly more time standing and walking and less time foraging than average, with low-ranking animals foraging more and laying less [87]. Since dominance hierarchy in our study group was not linear, we did not test the effect of a dominance rank on the probability of behaviour. However, differences between social clusters existed, as discussed. Low-ranking individuals from Cluster 3 also spent less time on locomotion, social and vigilant behaviour, but there were no differences in other behaviours between social clusters. Significant reduction of locomotion in this case could serve as an important indicator of the restricted ability to move for subordinate individuals.

**Behavioural synchrony.** According to our predictions, some horse dyads showed a higher index of behavioural synchrony than expected by chance. Behavioural synchrony is essential for maintaining group cohesion, which could be advantageous for the group members in terms of foraging efficiency, anti-predator strategies, etc. [76,88]. While an absolute group synchrony (all individuals engaging in one activity simultaneously) may be restricted by habitat, different physiological needs across age-sex classes, and other factors [89,90,91], behavioural synchrony within a dyad even with the absence of interactions could indicate strong social bond like was documented in bottlenose dolphin (*Tursiops aduncus*) males participating in synchronous surfacing [92]. In a population of > 150 feral horses in Portugal, animals synchronize strongly within distinct units (characterized as individuals that stayed closer than 15.5 m over 70% of the time) but also at an inter-unit, or whole-population level [93]. Similarly, in our study group, synchrony within social clusters was higher than between. In post-reintroduced Przewalski’s horses, high synchronization for the whole group (mean 87–91% for the mares and 87% for the mares with the harem stallion) was reported [94]. High behavioural synchrony within dyads of preferred associates indicates a strong social bond, consistent throughout different activities, meaning that horses deliberately choose their social partners.

**Sociality effect on space use.** We found that the areas with important resources were used unevenly by horses with subordinate individuals from Cluster 3 spending significantly less time there. No studies on the effect of individual social characteristics on space use throughout a day in bachelor males were performed. However, a study of captive American bison reported that high-ranking individuals preferentially used the barn and sand mound (critical resources), and low-raking individuals were left to use lower-quality sites [87]. Our findings highlight the importance of social relationship considerations when providing animals with extra resources in captive settings.

**Season effect.** While studying the impact of the season on behaviour of Przewalski’s horses was not a primary goal of this research, we acknowledge its possible effect. Firstly, the activity of horses peaked in the early morning and evening hours, when the temperatures were cooler. Care should be taken when comparing the rates of aggressive and/or affiliative interactions with other studies as the season may differ. An interesting direction of future studies would be the change of social structure between seasons.

**Welfare considerations and recommendations.** Time budgets of the study group were similar to those of feral individuals, and therefore no welfare violations were indicated. While space use was determined to be very unequal between different zones in the enclosure and depended on critical resources availability, it may be suggested to distribute those resources to different places within an enclosure to ensure social equity in its use. One of the main findings of this study was the observation of the formation of distinct social subunits within a group of males; hence, it would be interesting to study the formation of these units in the wild. The limited size of the enclosure itself could have made an impact on horses’ behaviour. Providing additional space for the entire group may positively affect the welfare of subordinate individuals and therefore should be considered by animal keepers. 

**Limitations of this study and future directions.** We acknowledge a comparatively small sample size used in this study, and therefore the results should not be extrapolated to the broad population. Unfortunately, the availability of specimens of the endangered species is limited. However, the use of permutations in the social network analysis helped to reduce the risk of type II errors by resampling the observed data multiple times and therefore increasing the power of statistical tests (comparisons made between original and randomized data for individuals rather than comparing the individuals across themselves).

Most of our studied animals were aged. While similar age did not significantly affect social preferences, the results should be interpreted with caution when comparing to wild and captive populations with younger individuals.

The interactions of Przewalski’s with Mongolian wild ass living at the same enclosure were not assessed; however, it would be an interesting direction for future studies. Historical records indicate that both species did co-occur in the past (before takhi extinction), and current ranges of reintroduced Przewalski’s horses and native khulans intersect [95,96].

## 5. Conclusions

We show that older bachelor male Przewalski’s horses form clear social preferences, associating and interacting in an affiliative way more with certain individuals than others. Interestingly, genetic relatedness and similar age did not appear to influence these social bonds. Instead, factors like similar levels of sociability and dominance played a more important role. Foraging and resting dominated the daily activities of all horses, although some individuals spent more time grazing (subordinates) while others relied more on provisioned hay. When a second enclosure was made available, all horses rested more and were less alert and used the second enclosure in a manner expected of ideal-free distribution. We did, however, observe unequal use of space within enclosures, with some individuals over-using areas rich in key resources like hay and water. Social structure had a strong influence on the horses’ daily activities and space use, as those that formed closer bonds tended to have similar activity budgets, synchronize their behaviour over all activities, and use the territory similarly. The group naturally divided into three subgroups (clusters) of two to four individuals based on spatial associations, affiliative interactions, and similarity in time budgets. Behavioural synchrony within clusters was higher than across them. One subgroup, consisting of two subordinate individuals, stood out by displaying less movement, social interaction, and vigilance, and by grazing more while feeding on hay less. This subgroup was also restricted in its access to resource-rich areas by the rest of the group. These findings have implications for management, highlighting the need to ensure equitable access to resources for all individuals, regardless of their social standing, as social constructs will emerge to influence resource use within bachelor groups of Przewalski’s horses.

## Figures and Tables

**Figure 1 animals-15-00053-f001:**
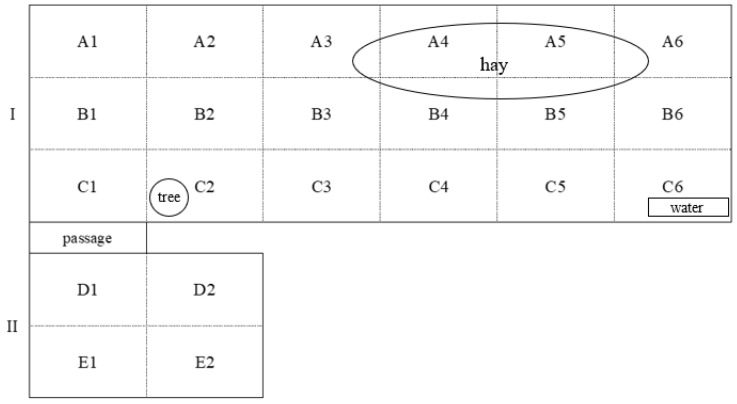
Enclosure map (I, II) for adult male Przewalski’s horses, Askania-Nova Biosphere Reserve (21 July to 14 August 2015).

**Figure 2 animals-15-00053-f002:**
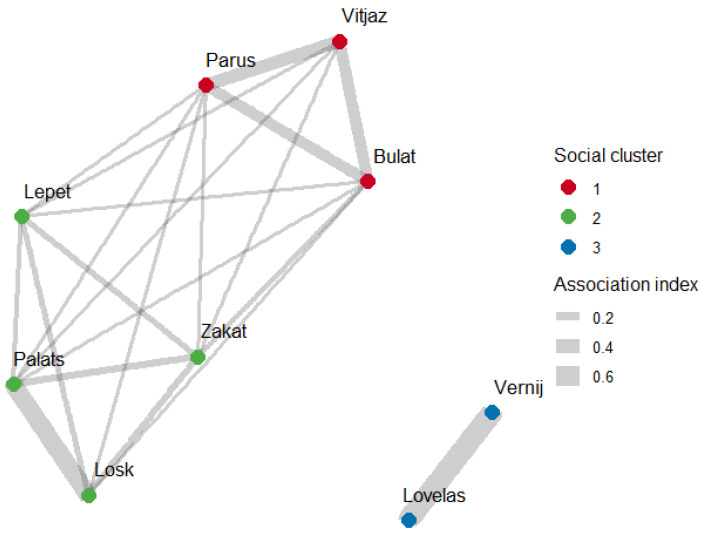
The social network of association indices in the bachelor group of Przewalski’s horses (Social cluster = communities detected by Louvain method).

**Figure 3 animals-15-00053-f003:**
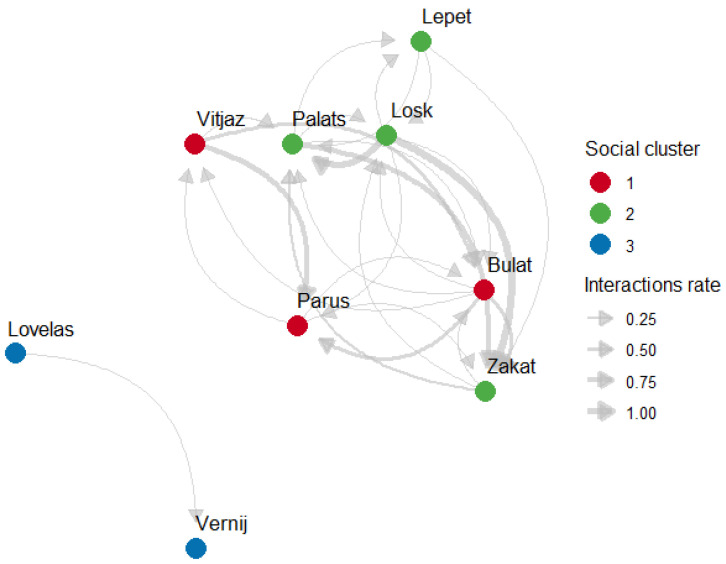
The network of rates of affiliative interactions in a bachelor group of Przewalski’s horses (arrow direction reflects initiator and a recipient of affiliative behaviour; Social cluster = communities detected by Infomap method).

**Figure 4 animals-15-00053-f004:**
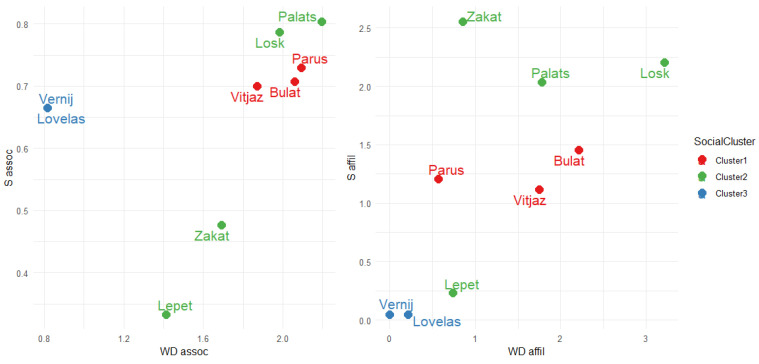
The relationships between weighted degree and strength derived from association network (**left**) and network of affiliative interaction rates (**right**) of Przewalski’s horses males.

**Figure 5 animals-15-00053-f005:**
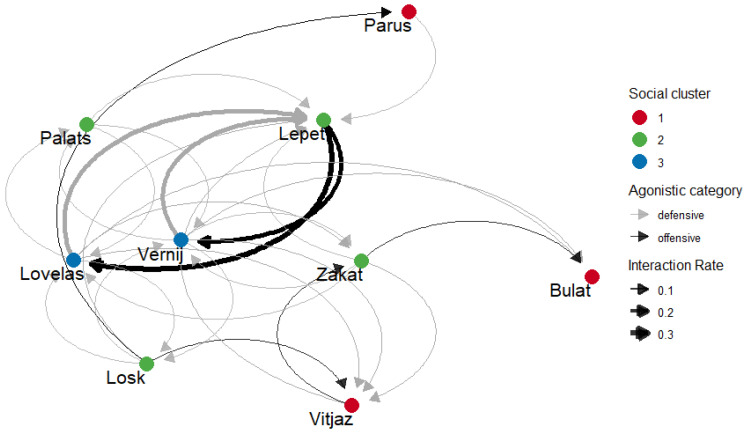
The network of rates of agonistic interactions in a bachelor group of Przewalski’s horses (arrow direction reflects initiator and a recipient of either offensive or defensive behaviour; weight = rates of interactions, clusters = communities detected by Infomap method).

**Figure 6 animals-15-00053-f006:**
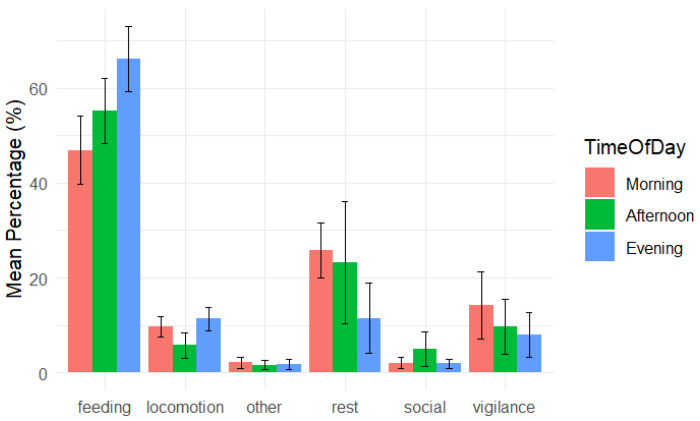
Daily time budget of Przewalski’s bachelor males by time of day (mean percentage of behaviour ± s.d.).

**Figure 7 animals-15-00053-f007:**
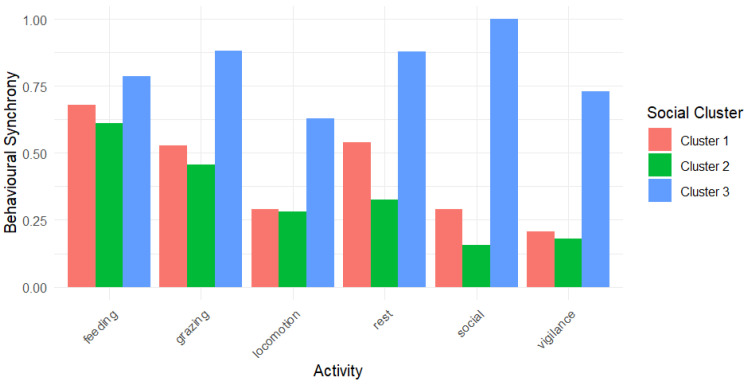
Behavioural synchrony in the bachelor group of Przewalski’s horses by activities by social clusters (see Table 8).

**Table 1 animals-15-00053-t001:** Behavioural subjects of adult male Przewalski’s horses, Askania-Nova Biosphere Reserve (21 July to 14 August 2015).

Studbook ID	Horse Name	Sire Studbook ID	Sire Name	Dam Studbook ID	Dam Name	Date of Birth	Age in 2015
5401	Zakat	3714	Zambar	2765	Vyshnya	25.05.2009	6 y
3756	Lepet	1447	Los	1577	Vesna	25.05.2002	13 y
3502	Palats	1447	Los	1864	Parussa	21.04.2001	14 y
3099	Bulat	1608	Lar	1382	Bulka	21.04.2001	14 y
3521	Lovelas	1447	Los	2516	Vesta	12.05.2001	14 y
3093	Vernij	1447	Los	2516	Vesta	06.06.1997	17 y
2906	Vitjaz	1608	Lar	601	Vira	03.05.1996	19 y
2935	Losk	1608	Lar	490	Vetka	27.05.1996	19 y
2665	Parus	1231	Volokh	1864	Parussa	11.05.1994	21 y

**Table 2 animals-15-00053-t002:** Ethogram of behaviours observed for male Przewalski’s horses housed away from females, Askania-Nova Biosphere Reserve (21 July to 14 August 2015). Ethogram based on [29,51].

Behavioural Category	Activity Pattern	Description
Foraging	drink	drinking water from a bowl
	feed on hay	eating hay from a hay pile, not separated by more than 10 s of rest, locomotion or other behaviour
	graze	eating grasses and herbs, not separated by more than 10 s of rest, locomotion or other behaviour
Rest	lay	laying relaxed
	head-to-tail	standing relaxed in an anti-parallel position with other individual
	stand	standing relaxed
Locomotion	gallop	running in a very fast four-beat gait
	trot	running in a fast two-beat gait
	walk	walking in a slow four-beat gait
Other	scratch	rubbing the body over the object (gate, fence, tree etc.)
	wallow	wallowing in the sand or on the ground
Social	mutual grooming	see Table 3
Vigilance	observe	standing with head and neck erect, ears erect and directed towards the object of interest

**Table 3 animals-15-00053-t003:** Social behaviours noted in a group of male Przewalski’s horses, Askania-Nova Biosphere Reserve (21 July to 14 August 2015).

Social Behaviour	Description
**Affiliative**	
Mutual grooming	two individuals standing beside each other, usually head-to-shoulder or head-to-tail, grooming (each) other’s neck, mane, rump, or tail by gentle nipping, nuzzling or rubbing
Mutual sniffing	the olfactory investigation involves sniffing various parts of another horse’s head and/or body
Rubbing	rubbing the body on the other horse’s body
**Agonistic**	
*Aggressive (offensive)*	
Bite	opening and rapid closing of the jaws with the teeth grasping another horse. The ears are pinned, and lips are retracted.
Bite threat	no contact is made. The neck is stretched and ears pinned back as the head swings toward the target horse
Ear threat	ears pressed caudally against the head and neck
Harassment	one horse pursuing another, usually at a gallop. The chaser typically pins the ears, exposes the teeth, and bites at the pursued horse’s rump and tail. The horse being chased may kick out defensively with both rear legs.
Kick threat	similar to a kick but without sufficient extension or force to make contact with the target. The hind leg(s) lifts slightly off the ground and under the body.
*Submissive (defensive)*	
Kick	one or both hind legs lifts off the ground and extend towards another horse, with apparent intent to make contact
Avoidance	a movement that maintains or increases an individual’s distance from an approaching horse or a horse initiating some behaviour. The head is usually low, and the ears are turned back.

**Table 4 animals-15-00053-t004:** Description of methods of social data analysis.

Measures	Associations	Affiliative Interactions	Agonistic Interactions
(1) Relationship metrics	Association index (AI) for each dyad was calculated as “joint occurrences,” i.e., the proportion of time individuals spent associated at a distance of 1 HBL out of all scan samples [69].	Rates of interactions for each dyad were calculated as the number of affiliative interactions in the dyad per hour.	Rates of interactions for each dyad were calculated as the number of agonistic interactions in the dyad per hour.
(2) Adjacency matrix construction	An undirected weighted square matrix constructed from the dyadic AI.	A directed weighted square matrix constructed from the dyad interaction rates.	A directed weighted square matrix con-structed from the dyad interaction rates.
(3) Social network construction	Social networks were constructed from adjacency matrices using a package *Igraph.*		
(4) Randomized social networks	For assessing non-randomness of associations and interactions data-stream permutations (*n* = 1000) were performed for each of three adjacency matrices [69,70]. In every randomization two random individuals associating (or interacting) were permuted between two dyads with swaps restricted to the same day of observation. Original data were replaced with randomized and for every permutation a new test statistic was calculated. A distribution test statistic in permuted data sets was then compared to observed measures to calculate *p*-values [69,70].		
(5) Community detection	Louvain algorithm (package *Igraph*)	Infomap algorithm (package *Igraph*)	NA
(6) Assortment	Assortment (propensity to associate or interact with a similar individual) within social networks was calculated with the function assortment.continuous from the *Assortnet* package (with package *SNA* used for node permutations).		NA
(7) Gregariousness (a) Node strength	Total proportion of time associated with anybody: the sum of all dyadic AI of a focal node (S_assoc_)	Total active sociability: the sum of both initiated and received interaction rates (S_affil_)	NA
(b) Weighted degree	Total proportion of time associated with anybody accounted for the importance of ties relative to their numbers and weights (WD_assoc_)	Total sum of both initiated and received interaction rates accounted for the importance of ties relative to their numbers and weights (WD_affil_)	
	The higher the weighted degree, the more contacts the total strength of the node is distributed to. For both node strength and weighted degree calculations a function degree_w from package *tnet* was used. The formula for calculations is in Appendix A (Formula (A1))		

**Table 5 animals-15-00053-t005:** Association indices between stallions of Przewalski’s horses within a bachelor group, calculated from spatial proximity data (Bold and “*” indicates preferred associates, *p* ≤ 0.05 when comparing with permuted networks; “+” indicates avoided associates, *p* ≥ 0.95 when comparing with permuted networks).

	Lepet	Losk	Lovelas	Palats	Parus	Vernij	Vitjaz	Zakat
Bulat	0.012+	0.005 +	0.000 +	0.002+	**0.325 ***	0.000 +	**0.305 ***	0.057
Lepet		0.107	0.000 +	0.082	0.012 +	0.000 +	0.012 +	**0.107 ***
Losk			0.000 +	**0.529 ***	0.007 +	0.000 +	0.000 +	0.139
Lovelas				0.000+	0.000 +	**0.665 ***	0.000 +	0.000 +
Palats					0.015 +	0.000 +	0.020 +	**0.156 ***
Parus						0.000 +	**0.357 ***	0.012 +
Vernij							0.000 +	0.000 +
Vitjaz								0.005 +

**Table 6 animals-15-00053-t006:** Rates of affiliative interactions between stallions of Przewalski’s horses within a bachelor group (Bold and “*” indicates preferred interactants, *p* ≤ 0.05 when comparing with permuted networks; “+” indicates avoided interactants, *p* ≥ 0.95 when comparing with permuted networks).

	Bulat	Lepet	Losk	Lovelas	Palats	Parus	Vernij	Vitjaz	Zakat
Bulat		0.000 +	0.031	0.000 +	0.092	**0.415 ***	0.000 +	**0.062 ***	**0.385 ***
Lepet	0.000 +		0.015	0.000 +	0.062 +	0.000 +	0.000 +	0.000 +	0.108 +
Losk	0.031 +	0.015 +		0.000 +	**0.923 ***	0.031 +	0.000 +	0.000 +	**1.077 ***
Lovelas	0.000 +	0.000 +	0.000 +		0.000 +	0.000 +	**0.046 ***	0.000 +	0.000 +
Palats	0.031 +	**0.031 ***	0.015 +	0.000 +		0.000 +	0.000 +	0.000 +	**0.723 ***
Parus	**0.046 ***	0.000 +	0.000 +	0.000 +	0.000 +		0.000 +	**0.031 ***	**0.031 ***
Vernij	0.000 +	0.000 +	0.000 +	0.000 +	0.000 +	0.000 +		0.000 +	0.000 +
Vitjaz	**0.354 ***	0.000 +	0.000 +	0.000 +	0.015 +	**0.662 ***	0.000 +		0.000 +
Zakat	0.015	0.000 +	0.077 +	0.000 +	**0.154 ***	0.000 +	0.000 +	0.000 +	

**Table 7 animals-15-00053-t007:** Individual sociability measures of stallions in a Przewalski’s horses bachelor group.

	S_assoc_	WD_assoc_	S_affil_	WD_affil_	DI
Bulat	0.707	2.060	1.453	2.213	1.178
Lepet	0.333	1.412	0.229	0.742	11.964
Losk	0.787	1.983	2.203	3.213	−0.333
Lovelas	0.665	0.815	0.046	0.214	−6.540
Palats	0.804	2.196	2.035	1.784	−2.792
Parus	0.730	2.092	1.209	0.567	2.031
Vernij	0.665	0.815	0.046	0.000	−5.883
Vitjaz	0.700	1.870	1.117	1.754	1.167
Zakat	0.476	1.691	2.555	0.857	−0.792

S_assoc_—node strength from associations matrix; WD_assoc_—node weighted degree from associations matrix; S_affil_—node strength from affiliative interactions matrix; WD_affil_—node weighted degree from affiliative interactions matrix; DI—dominance index (details on computations in Section 2).

**Table 8 animals-15-00053-t008:** Results of generalized linear mixed model examining differences of each activity patterns in relation to cluster membership (Cluster 1, Cluster 2, Cluster 3), time of day (Morning, Afternoon, Evening), availability of the enclosures (One enclosure available, Two enclosures available), and presence of the hay (Hay, No hay).

	Foraging		Rest		Locomotion		Social		Vigilance	
	*z*	*p*	*z*	*p*	*z*	*p*	*z*	*p*	*z*	*p*
(Intercept)	1.309	0.190	**−7.236**	**<0.01**	−19.236	**<0.01**	−8.043	**<0.01**	**−7.763**	**<0.01**
Cluster2	0.500	0.617	−1.110	0.267	−0.564	0.573	0.543	0.587	−0.332	0.740
Cluster3	−0.139	0.890	1.249	0.212	**−4.964**	**<0.01**	**−2.792**	**<0.01**	**−2.140**	**0.032**
Evening	**5.911**	**<0.01**	**−9.182**	**<0.01**	**3.918**	**<0.01**	**−4.691**	**<0.01**	**−2.564**	**0.010**
Morning	**−4.796**	**<0.01**	**2.387**	**0.017**	**2.802**	**0.005**	**−3.674**	**<0.01**	1.489	0.137
One enclosure available	**2.838**	**0.005**	**−6.577**	**<0.01**	1.146	0.252	−1.266	0.205	**5.878**	**<0.01**
No Hay	-	-	**5.691**	**<0.01**	**5.178**	**<0.01**	0.372	0.710	**5.433**	**<0.01**

**Bold** indicates that the given factor was a significant predictor in variation of probability of each activity pattern. The following variable levels are reference categories, reflected in the value of the intercept:Social Cluster = Cluster 1, Time of day = Afternoon, Availability of the enclosures = Two enclosures available.

**Table 9 animals-15-00053-t009:** Spread of participation index (SPI) of the space utilization in male Przewalski’s horses between two periods of enclosures availability.

Horse Name	Two Enclosures Available	One Enclosure Available
Bulat	0.489	0.448
Lepet	0.502	0.575
Losk	0.483	0.528
Lovelas	0.339	0.312
Palats	0.484	0.536
Parus	0.467	0.434
Vernij	0.342	0.297
Vitjaz	0.481	0.475
Zakat	0.493	0.515

**Table 10 animals-15-00053-t010:** Results of a generalized linear mixed model examining differences of feeding on hay and grazing in relation to cluster membership (Cluster 1, Cluster 2, Cluster 3), time of day (Morning, Afternoon, Evening) and availability of the enclosures (One enclosure available, Two enclosures available).

	Feeding on Hay		Grazing	
	*z*	*p*	*z*	*p*
(Intercept)	**−9.224**	**<0.01**	**−13.640**	**<0.01**
Cluster2	**3.201**	**0.001**	−2.269	0.023
Cluster3	**−13.464**	**<0.01**	**11.320**	**<0.01**
Evening	0.098	0.922	**6.642**	**<0.01**
Morning	**−10.328**	**<0.01**	**4.737**	**<0.01**
One enclosure available	**10.968**	**<0.01**	**−7.607**	**<0.01**

**Bold** indicates that the given factor was significant predictor in variation of probability of each activity pattern. The following variable levels are reference categories, reflected in the value of the intercept: Social Cluster = Cluster 1, Time of day = Afternoon, Availability of the enclosures = Two enclosures available.

**Table 11 animals-15-00053-t011:** Dyadic behavioural synchrony indices (BSI) for the bachelor group of Przewalski’s horses computed as the mean over all activities (Bold and “*” indicates BSI higher than expected by chance, *p* ≤ 0.05 when comparing with permuted networks).

	Lepet	Losk	Lovelas	Palats	Parus	Vernij	Vitjaz	Zakat
Bulat	**0.430 ***	**0.391 ***	0.205	**0.406 ***	**0.607 ***	0.204	**0.579 ***	**0.390 ***
Lepet		**0.485 ***	0.197	**0.499 ***	**0.404 ***	0.196	**0.403 ***	**0.445 ***
Losk			0.213	**0.709 ***	**0.406 ***	0.208	0.336	**0.553 ***
Lovelas				0.215	0.219	**0.881 ***	0.176	0.174
Palats					**0.465 ***	0.212	**0.373 ***	**0.549 ***
Parus						0.23	**0.596 ***	**0.397 ***
Vernij							0.186	0.169
Vitjaz								**0.349 ***

**Table 12 animals-15-00053-t012:** Mean behavioural synchrony indices (BSI) in the bachelor group of Przewalski’s horses for different activities for social clusters, within and between clusters, and the difference between those and a *p*-value derived from the comparison of observed and permuted differences.

Activity	Mean ± sd	Cluster 1	Cluster 2	Cluster 3	Mean BSI Within a Cluster	Mean BSI Between Clusters	Mean BSI Difference	*p*-Value
Grazing	0.322 ± 0.185	0.528	0.457	0.881	0.521	0.246	0.275	0.000
Feeding on hay	0.316 ± 0.275	0.680	0.610	0.786	0.649	0.188	0.461	0.001
Locomotion	0.177 ± 0.126	0.290	0.280	0.629	0.318	0.122	0.196	0.000
Rest	0.256 ± 0.171	0.539	0.327	0.878	0.446	0.183	0.263	0.000
Social	0.090 ± 0.182	0.291	0.156	1.000	0.281	0.017	0.264	0.001
Vigilance	0.130 ± 0.123	0.206	0.180	0.730	0.243	0.087	0.155	0.002

## Data Availability

Data available upon request.

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
