# Peer review of "Social Relationships of Captive Bachelor Przewalski’s Horses and Their Effect on Daily Activity and Space Use"

_animals, 2024, doi:10.3390/ani15010053_

Round 1

Reviewer 1 Report

Comments and Suggestions for Authors

It was a very nice, well-written, and interesting article to read. I congratulate the authors for such a nice piece of work. I have only one central reservation with your study. Is the sample size (n=9) enough to conclude the results? Secondly, please recheck the entire manuscript and see missing references. I have pointed out a few below.

Lines 76-77, 81-82, 88-89, 92-93: Please add the references.

Line 93: Please add further text on how these factors can affect breeding on your studied breed.

Line 263-265: Can the season of the year also affect horse behavior? Should it be taken as fixed effects? I know you have not studied this variable — just a thought.

Author Response

Please see file attached.

Reviewer 2 Report

Comments and Suggestions for Authors

The manuscript entitled Social relationships in captive bachelor Przewalski’s horses and their effect on time budgets and space use focuses on the behavioural analyses of nine Przewalski’s horse stallions sharing the same small enclosure in Askaniya-Nova Nature Reserve. The authors stated that the motivation behind the study is to describe natural behaviour in horse bachelor groups and, through a better understanding of the individual differences between the animals and assessing their group behaviour, to select the best candidates for reintroduction to the wild.

Although the idea is interesting and potentially valuable, the study lacks an adequate sample size (all the animals in the focal group are over the age recommended for reintroductions) and a sufficient amount of data to be tested under relevant conditions. The so-called bachelor group is formed out of nine animals, but except one, all of them are older than 13 y (some over 20), which is highly improbable to be found in nature. Moreover, even if its mentioned in the text how important may be the mating experience for the stallion's later behaviour, the authors do not show any information about it nor count it for the statistics. Statistics is the second part of the paper, which is of limited explanatory value as all the models are based on very minimal data  (for example, 107 agonistic interactions only, most of them provoked by one animal) collected for a very short and specific time period (peak summer with very high temperatures, which the authors did not address properly in the discussion).

Unless the data collection would significantly extend in range, the results are unconvincing and, therefore, not ready to be published.

Specific comments:

L: 26-28: The meaning of those sentences is unclear; please consider rewording.

L:30: Why do you think the bachelor groups are less studied than other types of horse groups? (please add references)

L: 35, 36: That's a notoriously known fact that captive animals need resources to be spatially distributed so the dominant ones are not bullying the submissive ones while eating.

L:37: Health and temperament were not tested in the study; therefore irrelevant here.

L:43: „The natural social behaviour of group-living species in captivity:“ Natural behaviour in captivity depends on given conditions. Do you expect the horses to behave naturally in an artificially selected group in a small environment with limited access to resources?

L:48: Add the species

L:54: Instead of highly successful, more relevant just to inform the reader that there are such projects.

L:59: Add a current number

L:65: „Post-release social species“has an unclear meaning; please reword.

L: 75: Citation missing

L:76: Confusing to talk about „feral“ horses in this context, it is better to use domestic horses.

L:84: Skip „we“

L:94, 95: Please put in logical order.

Several shortcomings in Methods: unclear if there were khulans or onagers in the enclosure, their effect on horses' behaviour not commented, horse characteristics missing: mating experience, health state, condition, how long the group lasts, relatedness, unclear if there was a grass for grazing or not, the size of the water source not given, shelter: yes/no, where, how it looks, for how many animals? How many observers participated? How was the scoring of 1 horse length when several individuals were close? How were the hours/day divided in order to calculate the time budget (since when to when was morning etc., how was the observation distributed?)

L:209: Descriptive statistics missing, how many interactions observed in total, for each horse…

L:279: Figure 2, explain AI

L:294: Unclear formulation

L:320: The values are missing for the test.

L:341: Table 5: The caption has to include all the abbreviations.

L:361: To interpret highly insignificant results as subtle positive is confusing.

Discussion: completely missing points about the weather effect on behaviour (high temperatures – limited reactivity of the horses, senescence (three horses 19+), the very limited size of the enclosure, presence of other animals. And finally, how are the results relevant for the selection of horses for reintroduction projects, considering the age of the horses.

Reviewer 3 Report

Comments and Suggestions for Authors

Thank you for the opportunity to review this manuscript. Social network analysis and space use in Przewalski horses are both current topics and of interest to me, along with a very broad audience of scientists. This paper adds to a growing body of research on behavior, conservation, and management of this once endangered species. A strength is the enlightening comparison of social networks created by proximity (association) and behavioral (affiliation) data. The analyses also revealed different types of social groupings among stallions, from tightly bound brothers to loosely organized bands. Anyone familiar with herd dynamics will have observed similar patterns, but in this case the observations are supported by empirical data. The closing paragraph of the manuscript offers an excellent synopsis of key findings from the study.

The study has merit and should be published. The manuscript, however, has several shortcomings that I believe need to be addressed before publication. My strongest recommendation is that the authors submit a more streamlined version of the manuscript that focuses on the social network analyses and omits secondary questions and findings. For example, the additional pasture area created some effect on time budgets, but the ideal free distribution model and analysis did not add anything meaningful to this study and, aside from mentioning and perhaps adjusting for the change in area, I suggest dropping it from the paper.

I have listed several additional points for the authors to consider as they revise the paper, generally accompanied by a reference to specific parts of the manuscript. I hope they are helpful.

1)    The analyses were based on a small sample size of nine equids, relatively few observations (65 hours) made over a limited period of time (5 weeks). A consequence is low power that probably affected some analyses along with limited generalizability of the results to other herds, locations, seasons, etc. No acknowledgement of this limitation is made in the paper.

It is likely that some non-significant results were affected by power but in some cases they are presented as meaningful trends in the report. One example is the correlation between genetic relatedness and social association.

·        Section 3.1.15 addresses genetic relatedness and affiliation. This will be important to many readers. The authors report a series of weak, non-significant correlations.

·        Lines 354-358. Both association and affiliative behavior had small positive, but non-significant relationships with genetic relatedness. The term ‘slightly’ is used to describe the correlation, and the sentence that begins with ‘in contrast’ compares results with nearly identical statistics.

A second example is in the analysis of behavioral synchronization. Including a synchronization index (Line 245) was a current and valuable part of the analysis.

·        Line 428. Behavioral synchrony overall was not significant. The authors, however, over-report this non-significant finding as horses ‘showing some behavioural synchrony’. A more conservative interpretation would be appropriate. 

2)    Social network analysis is the most valuable part of this study. Surprisingly, the expected social network metrics, i.e., betweenness, degree, centrality, and strength, are not reported. I suggest that the authors both provide a rationale for applying different metrics and include the more traditional network analysis metrics, perhaps in a table or Appendix.

·        Line 242. The equations may be challenging for some interested readers. The model is easier to understand when the terms in the equation are defined in verbally. In some cases, as on line 242, the definition is missing or had been presented previously, which could make it difficult for reader. A table with the equations and definitions might be simple and helpful. 

3)    In the Methods section, the authors note that the takhi shared the enclosure with 18 onagers. Nothing further is mentioned about it.

·        As a minor point, the name khulan is first used (Line 164) then followed by a note that it is an onager subspecies (Lines 174-175). For readers unfamiliar with onagers I recommend adding the common name ‘Mongolian wild ass’ and using the species or common name thereafter.

·        As a major point, I assume that the two species cohabitated and shared resources over the 5-week observation period. Do these species co-occur naturally? Did the onager and horses interact in any meaningful way that would affect resource access and spatial distribution of the horses? Did the researchers observe the onager space and resource use? 

4)    Additional information is needed in the Methods section about behavior sampling.

·        Lines 184-185. The authors report making observations for 25 days, from 0500h – 2030 h for a total of 65 contact hours. More detail is needed about the observation schedule. Were takhi observed every day? How many hours were they observed each day and at what times? Were observation hours balanced across morning, mid-day, and evening? Enough information should be provided to demonstrate an unbiased sampling across time.

·        On Line 371 the authors first note that horses spend time grazing. This should, but was not, included in the Methods section (see Lines 174-177). Did all areas have enough browse for grazing? How much of their diet was browse compared to the provisioned hay? Free grazing could change the value of the hay as a resource and affect space use.

·        A range of values and their interpretation is needed for the SPI in order to make sense of the reported index Lines 252-262. of .45. The same is true for the Electivity index (Line 253)

·        Lines 540-543. The sentence refers to the focal horses being part of an ‘established group’. A citation is needed to support the assertion that there is reduced agonism in established groups. How long the focal horses had been housed together should also be in the Methods section. 

5)    The Result section appropriately emphasizes the network analyses. They are based on behaviors listed in the ethograms, but only a limited number of summary statistics of behaviors and space use are reported in the manuscript. The time budgets reported in section 3.2 and Figure 6 are a great start. Further breakdown of the descriptive information would be valuable to many readers and could appear in an Appendix. (For example, how common were kick threats relative to other aggressive behaviors? How much time did horses spend in quadrants with hay?)

6)    Line 342 section 3.1.4. This section on agonistic social behavior feels unfinished and I am left wanting more. It does not, for example, address apparent individual differences in agonistic behavior. In addition, aggressive and submissive behaviors were combined in the analysis, yet they might be opposite in directionality.

·        Perhaps the authors could use total interactions if the agonistic interaction results are not enlightening. As in this study, other published studies have found that affiliative interactions are more common than agonistic interactions, and some report that combining affiliative and agonistic (total interactions) does not change the overall results (e.g., https://doi.org/10.1016/j.anbehav.2024.08.009).

7)    The Discussion section reviews the results of the study well but needs further development.

·        While the authors report the main findings, they rarely address limitations to the study, alternative explanations, or suggestions for future research. One example (Lines 592-595) is in the discussion of behavioral synchrony. While social connections between horses might best explain greater synchrony within clusters (and was statistically significant), circadian patterns and weather conditions, among other factors, may also contribute to synchronous behavior patterns. These explanations are not mentioned in the paper.

·        Lines 545-563. This Discussion paragraph compares time budgets across a range of studies. The comparison is, of course, important but reporting specific statistical values, e.g., percent and standard deviation, is not necessary and a verbal discussion comparing trends would be sufficient.

Comments on the Quality of English Language

Numerous minor errors appear throughout the text, including spacing, punctuation, missing words, long sentences without breaks, and others. Some may be due to writing in English, but they affect the overall quality of the paper. Closer proofreading is needed to create a more polished and professional report.

I have listed several examples from the text.

·        Lines 313 and 313. One states (Figure 2) and the other states (see Figure 3). Using consistent format and style would result in a more polished report.

·        Line 347 has an oddly placed dash that does not help organize information in that sentence; starting a new sentence would be grammatically preferred and easier to read.

·        Line 127: Incorrect use of semicolon creates a run on sentence. Breaking it up into a few sentences can help improve readerability.

·        Lines 143-144: awkward sentence grammatically affects meaning; it may be simply missing ‘and’, i.e.,…’and demonstrate’.

·        Lines 337-339. In this paragraph authors refer to interactions and then to interactants. The point of this paragraph is sociability, but the result point is unclear.

·        Lines 389-391: The point is clear, but the list of items would improved by applying parallel grammatical construction.

·        Line 405. The sentence uses the term ‘subunit’ but I believe is referring to ‘cluster’ as noted in the table. Please use the same terminology throughout for clarity.

·        Line 505. What is a type 2 subgroup? This term was not previously used in the paper.

·        Line 547: The line after the colon begins with “: reports” but something—a reference?—appears to be missing.

I also found formatting irregularities and insufficient information in the Figures and Tables. I have listed them here.

·        Tables 4 and 5 both show arrays of relationship but are structured differently. The array in Table 6 has redundancies but is symmetrical. The Table 6 format is much easier for evaluating relationships between one horse and others by scrolling down just one column or across one row.

·        Figure 3 (and Figure 5). The Infomap method of affiliative interactions indicates strength with line width. The arrows suggests sender-receive direction, but this is not explicitly clear from the written text. Section 3.1.3 addresses directionality but does not refer to the arrow directions in Figure 3, and thus does not clarify the issue.

·        Table 5. The column headers reflect important indexes, but the reader may not recall what the symbols refer to. I suggest including the symbols and definitions in the table caption or in the paragraph connected to the table (it is too much to ask the reader to comb through the rest of the text, as noted in the parentheses).

·        Figure 6 and Lines 376-377. The time budget is a valuable addition to the results, but more is needed in the methods to support the analysis. What time of day (range) was classified as morning, afternoon, and evening? How many observation hours were made at each time period? Was the observation sampling balanced over the five weeks? As horses are active 24 h, I suggest that the authors mention what hours are absent from the data set.

·        Line 384 and Figure 6. Significant differences should be marked on the figure with * and noted in the caption.

·        Tables 7 and 8 report results for the GLM. Column 1 appears to be missing cluster 1, afternoon, second enclosure, and hay. I may be confused about this analysis but, if so, other readers will likely be confused as well. I suggest that the authors explain, especially since the written text refers to these variables.

Reviewer 4 Report

Comments and Suggestions for Authors

Overall:  Overall, the paper adds a lot to our knowledge on social interactions between individual horses and how social preferences influence time and resource budgets. 

Abstract:  Overall the abstract and Simple Summary do a good job of providing a summary of the background for this study and justifying the need along with presenting an overview of findings and applications of findings. 

Introduction:  Overall, the introduction does a very good job of introducing the history of the Przewalski’s horse and the need for more information about social behaviors for better introduction into the wild or support in captivity.  The authors do a very good job of explaining the existing literature and need for this study.

Line 129 – 130 :  There is more evidence that dominant hierarchy does not actually exist in natural horse herds and that agonistic behaviors are more about space rather than any kind of hierarchy.  There is little evidence outside of domestic settings that a hierarchy is present and that the “dominant” stallion, as mentioned by the author, is only named such due to breeding status rather than a specific social rank.   Concepts of hierarchy in horses have been derived only from horses in limited enclosures with limited resources and only seems to be a reflection of extreme social preferences for sharing rather than a reflection of natural social hierarchy.  (See work by Rees, Maeda, and more recent publications from ISES in how dominance is not an accurate reflection of natural horse societies). 

Materials and Methods:  The materials and methods section does a very good job of providing detail with regards to the study location and setup.  Additional information is needed with regards to number and training of observers. 

Line 184 :  Specify number of observers at any given time.  Also be sure to specify overall number of observers that took part in this observation part of the experiment and the training that was provided to the observers. 

Results:  The authors do a great job of not only presenting the quantitative findings and supporting the findings with appropriate visuals, but they also combine these findings with individualized variations in behavior with regards to affiliative and agonistic interactions.  This helps provide additional depth to the research that is often lacking in other studies. 

Line 302:  “Preferred and avoided associates” is a great alternative to “dominant hierarchies” and should be included earlier in the paper to more accurately describe social interactions and social structures within the group. 

Line 342:  Based on previous mentions, the use of “dominance hierarchy” should be replaced with something that provides more descriptive approaches to social preferences rather than a hierarchy.   The authors do a good job of defining “dominant” and “submissive” behaviors which are descriptors of interactions between two individuals rather than a definitive social structure of the group. 

Discussion and Conclusion:  Overall, the authors do a good job of contextualizing their findings into existing literature and are very cognizant of the limitations of the findings given the artificial grouping in a man-made enclosure.  The authors also do a good job of discussing the potential impact of individual histories on behavior, something that is not often discussed in similar papers.

Using the term “dominant” should again be clearly defined, even in the discussion.  Due to more recent research, the concepts of dominance hierarchy can be seen to have equal relevance if the social structure is described as social preferences and behavioral interactions rather than a hierarchy of dominance.  The authors do a good job of describing the variations of findings in the social structure of similar papers so the likelihood of a non-linear social structure with individual variations in behavioral expressions and social preferences (without mares) is just as likely to exist as aligning it with concepts of social dominance theories.  As described earlier in the paper and in other papers, the term “dominance” can refer to the breeding male (not relevant in this paper) or a type of behavior (although agonistic or aggressive is more suitable).  It is recommended, therefore, that the authors consider rewording portions of the section aligning the findings with dominance hierarchies. 

Round 2

Reviewer 3 Report

Comments and Suggestions for Authors

An impressive amount of work went into this revision with thoughtful attention to reviewer feedback.

In reading the revision I noted a possible grammatical issue that could lead to misinterpretation on lines 935-936, with double negative phrasing. It should read 'not significant in either associations..... or interactions. Please check with the editor.

Assortment by age was not significant in neither associations (assortment coefficient, AC 935 = -0.084, p = 0.415), nor interactions (AC = 0.025, p = 0.249).